# Absence of fibroin H sequences and a significant divergence in the putative fibroin L homolog in *Neomicropteryx cornuta* (Micropterigidae) silk

Yume Imada [1], Lenka Rouhova [2,3], Martina Zurovcova [2], Miluse Hradilova [4], Sarka Podlahova [2,3], Hana Sehadova [2,3] & Michal Zurovec [2,3] ✉

Micropterigidae is regarded as the sister group of all the other Lepidoptera, providing important insights into the evolution of Lepidoptera. However, the gene and protein profiles of silk from Micropterigidae have not yet been identified. In this study, we investigate the components of silk cocoons of the micropterigid species *Neomicropteryx cornuta*. Here we show that the protein fibroin heavy chain (FibH) is absent in the silk of *N. cornuta* and that the putative homolog of fibroin light chain (FibL) is also absent or severely altered. This is confirmed by transcriptome and genome analyses of the conserved regions in this species. The examination of the synteny around the *fibH* genes in several Lepidoptera and Trichoptera species shows that the genomic region containing this gene is absent in another micropterigid species, *Micropterix aruncella*. In contrast, we found putative orthologs of *fibH* and *fibL* in the representative transcripts of another distinct clade, Eriocraniidae. This study shows that the loss of FibH and the loss or severe divergence of FibL occurrs specifically in the family Micropterigidae and reveals dynamic evolutionary changes in silk composition during the early evolution of Lepidoptera. It also shows that silk proteins without FibH can form a solid cocoon.

Silk is a water-insoluble proteinaceous material prominent for its bioadhesive properties, and is produced by various Chelicerata and Hexapoda species[1]. In insects, silk production has evolved independently at least 23 times[2]. Particularly, most larvae of Lepidoptera (moths and butterflies) and Trichoptera (caddisflies), constituting members of Amphiesmenoptera, synthesize silk via the salivary glands, known as silk glands (SG)[3,4]. Aquatic caddisfly larvae spin silk under water to construct capture nets, portable cases, and pupation cocoons. Moths primarily use silk for spinning cocoons and making portable cases and protective shelters.

Silk produced by moths and caddisflies comprises few proteins synthesized in the silk glands and stored in a liquid form, which hardens after spinning[5]. The primary components of silk filaments, heavy and light fibroin chains (FibH and FibL, respectively), are distributed among moths and caddisflies. These fibroin subunits form an insoluble fiber core that provides mechanical strength to the silk. The fibroin core is coated by several layers of soluble, adhesive proteins that bind the fibers[5,6]. Regarding the silkworm and honeycomb moth, *Galleria mellonella*, the adhesive layer can be formed by sericins, which are hydrophilic glue proteins[7,8], and caddisflies also incorporate several presumptive adhesive proteins (pseudofibroins or cadhezins)[9]. The silk of moths and caddisflies also contains several other proteins, such as proteases and antibacterial proteins[7,10].

Recent genome sequencing has demonstrated homology between several *fibH* and *fibL* genes in terrestrial moths and aquatic caddisflies[11–14]. The *fibH* gene is large, has short, conserved characteristic sequences at both ends, and variable repetitive sequences in its central region. Despite conserved regions at the N- and C- terminus, FibH molecules have diversified considerably in both groups[13,15]. In moths, FibH molecules are characterized by their hydrophobicity and motifs, which can form beta-sheet structures responsible for mechanical strength. In contrast, the FibH molecules of caddisflies are hydrophilic and larger, and mostly lack the amino acid residues required to form beta-sheet crystallites[12]. In contrast, the *fibL* gene is much smaller and does not contain repetitive sequences. The protein products of both the fibroin genes form dimers[16].

[1]Kyoto University, Graduate School of Science, Kyoto, 606-8502, Japan. [2]Biology Centre of the Czech Academy of Sciences, Institute of Entomology, Ceske Budejovice, 37005, Czech Republic. [3]Faculty of Science, University of South Bohemia, Ceske Budejovice, 37005, Czech Republic. [4]Institute of Molecular Genetics, Academy of Sciences of the Czech Republic, Praha, 142 20, Czech Republic. ✉e-mail: zurovec@entu.cas.cz

Although there has been progress in the study of silk proteins, the gene and protein profiles of silk from Micropterigidae, a group that probably split from other Lepidoptera over 300–200 million years ago, remain unidentified[17,18]. To fill this knowledge gap, we provide evolutionary insights into an important biomaterial. We used advanced omics to identify the silk-related genes and their secretory products in *N. cornuta*. In addition, we extended our search for fibroin genes to another micropterigid species and two species of Eriocraniidae of the family Eriocranoidea, representing a deep node in the non-ditrysian moth phylogeny. We show that the FibH is absent in the silk of *N. cornuta*, and that the putative homolog of the fibroin light chain (FibL) is either absent or significantly altered. Furthermore, we show that the remaining silk proteins are able to form a tight cocoon even in the absence of fibroins

## Results

### Morphology of SG and cocoons

To investigate the arrangement of the silk gland (SG) in *N. cornuta*, we performed micro-CT imaging (a technique that uses X-rays to view the inside of an organ layer by layer). Our 3D study revealed a lateral looping pattern in SG. In the larval model of *N. cornuta*, SG extended approximately to the middle of the body, terminating at the posterior level of the third abdominal segment (Fig. 1). The anterior part of SG, corresponding to what is typically known as the anterior silk gland (ASG) in the other moths and caddisflies, extends from the cephalic region to the end of the first thoracic segment (Fig. 1A). Subsequently, there was a distinct thickening of SG, continuing up to the first abdominal segment. This corresponds to the middle silk gland (MSG) in other lepidopterans. There was a thin layer of cells on the SG surface surrounding the lumen with stored silk (Fig. 1D). In the first abdominal segment, the SGs expand further and form two loops: a shorter loop toward the head and a longer loop toward the lateral part of the body (Fig. 1). This region may correspond to the posterior silk gland (PSG) observed in other lepidopterans. This was marked by large secretory cells encircling the spacious lumen (Fig. 1E).

The silk stored in the SG lumen showed notable color differences when stained using Masson's trichrome (Fig. 1 B–E). The silk material in the ASG appeared red, whereas that in the MSG and PSG appeared dark blue. These color differences suggest that silk undergoes structural changes as it moves through the SG. Notably, this staining method did not show the expected differentiation between the fibroin core and the coating layers within the lumen (Fig. 1).

The cocoon of *N. cornuta* (Fig. 2) was oval-shaped and ~4 mm long. It featured a thin, compact wall that formed an almost impermeable envelope. The cocoon's filaments, ~3 µm wide, were ribbon-like and tended to fuse with the adhesive layer, suggesting a high content of sticky material. Notably, the front part of the cocoon included chimney-shaped pores with a diameter of 40–50 µm (Fig. 2A–C). These perforations likely formed through intentional circular movements of the larval head during spinning, thereby enabling environmental interactions with the pupa.

### Detection of putative silk structural proteins

Transcriptome analysis was conducted to identify the major silk components of *N. cornuta*. First, we isolated RNA and prepared an RNA-seq cDNA library. Transcriptomes of the entire larvae were constructed using de novo and genome-guided transcriptome assembly methods. The de novo transcriptome served as a proxy protein database. Cocoon proteins were digested with trypsin and the resulting peptides were identified using peptide mass fingerprinting by comparing the experimental and predicted MS/MS spectra. We discovered 113 proteins, 62 containing a signal peptide (Table S1). As shown in Table S1, these proteins were categorized into four groups. The first group (13 members) encoded potentially large structural silk proteins characterized by repetitive sequences. The second group comprised 24 members that encoded various enzymes (Table S1). The third group comprised proteins with homologs in other species, but no established association with silk (nine members). The fourth group comprised 16

small proteins with unknown functions, some of which also exhibited repetitive sequences and likely contributed to the structural integrity of silk.

The composition of the *N. cornuta* cocoon was particularly notable in the first structural group. It comprised gene products organized into small clusters that encode putatively large structural silk components (Table 1). One of these clusters, located in the JAHKQU010000031.1 contig of the publicly available genome ASM2038319v1, included genes encoding sericin-like proteins (Srcl1, Srcl2, and Srcl3). These highly hydrophilic proteins had repetitive sequences with a significant proportion (21–57%) of serine residues. Srcl2 and Srcl3 were particularly large, with molecular weights of 933 and 1450 kDa, respectively. Additionally, four other genes in this group resembled caddisfly cadhezins[9]. These genes were encoded by two exons and arranged in pairs on two genomic contigs (JAHKQU010000021.1 and JAHKQU010000101.1), with their coding regions split into two exons. Their protein products lacked sequences that were capable of forming crystal domains and did not exhibit sequence conservation with fibroins or sericins. The structural group also comprised four *zonadhesin-like* genes on the JAHKQU010000031.1 contig that encode protein products between 29 and 130 kDa. These products displayed a high proportion of cysteine residues (14–15%) and conserved Til/EGF2 domains, suggesting their possible role in protease inhibition. Finally, the structural group also included two large genes (containing four and ten exons) encoding putative mucin-like proteins, Muc1 and Muc2, located as singletons on contigs JAHKQU010000032.1 and JAHKQU010000010.1, respectively. These proteins had hydrophilic, repetitive sequences with serine residues comprising 10 and 13% of the sequences, respectively.

### *N. cornuta* silk lacks a close homolog of FibH

Silk proteomic analysis revealed that proteins similar to FibH were absent in *N. cornuta*. Thus, we performed a comprehensive survey of the transcriptomic and genomic sequences of *N. cornuta* using the BLAST algorithm, with conserved regions of known *fibH* genes as reference points, in Lepidoptera and Trichoptera. Our analysis also included the genome sequence of another micropterigid species, *Micropterix aruncella*. However, we failed to identify *fibH*-like sequences. These results reinforce the fact that the *fibH* may have been entirely lost from the genomes of both micropterigid species, or it may have diverged substantially, rendering it unrecognizable by sequence similarity.

The conservation of the genomic region harboring the *fibH* gene was investigated based on genome sequence analyses of several lepidopteran and trichopteran species, and it was found that the *fibH* gene is typically located between two genes, *prospero* (*pros*) and *dihydrolipoyllysine residual succinyltransferase* (*DRSC*). This arrangement is conserved among many lepidopterans and trichopterans, ranging from, *Nematopogon swammerdamelus* (Adelidae), *Incurvaria masculella* (Incurvariidae) in Adeloidea, *Nymphalis io* (Nymphalidae), and *B. mori* (Bombycidae); additionally, in two caddisflies suborders, including *Hydropsyche tenuis* (Hydropsychidae) and *Parapsyche elsis* (Hydropsychidae) in Annulipalpia, and *Himalopsyche kuldschensis* (Rhyacophilidae) in Rhyacophiloidea. In addition, our study revealed the relatively conserved presence of homologs for the additional two genes, ubiquitously expressed transcript (*UXT*) and repetitive organellar protein (*ROP*), in the same genomic region, as shown in Fig. 3.

There were some minor rearrangements within the genomic region, such as the opposite orientation of the *UXT* gene between caddisflies and moths, an inversion of *DRSC* in the ditrysian Lepidoptera *B. mori* and *Nymphalis io*, and the translocation of *ROP* in *H. tenuis* and *P. elsis* (Fig. 3). Notable conservation of the entire region was observed in most of the Lepidoptera and Trichoptera species that were investigated. Our results also suggest that more extensive restructuring occurred in the two micropterigid genomes, as *UXT* and *ROP* relocated to different genomic regions, and the original DRSC copy was lost and replaced by an intronless copy elsewhere in the genome (contig JAHKQU010000032.1, in *N. cornuta*; chromosome 15 in *M. aruncella*). Although the 3′-terminal part of the *DRSC* gene remained

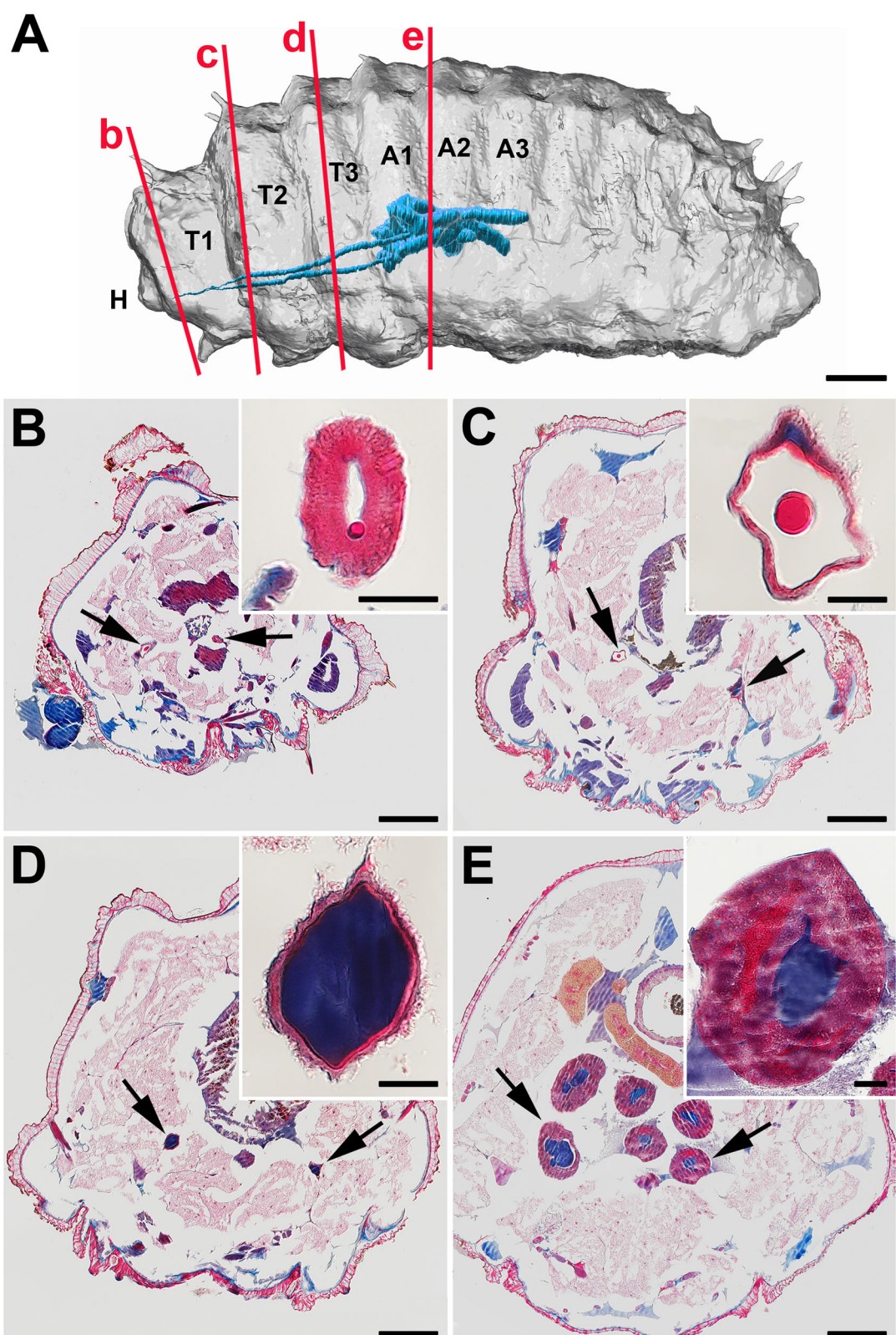

**Fig. 1 | Morphology and histology of the *N. cornuta* silk gland. A** 3D visualization of a final-instar larva of a micropterigid moth*, N. cornuta*. μ-CT images were optically sliced along the longitudinal axis and modeled with the Imaris software (Oxford Instruments). H - head, T1, T2, T3 – thoracic segments, A1, A2, A3 – first three abdominal segments. SG, indicated in blue, extends only up to the middle of the body. The red lines with the letters b–e correspond to the approximate position of cross sections (**B–E**). **B–E** Paraplast transverse sections through the larval body of the 5th instar stained with Masson trichrome stain (arrows show the SG). A highly magnified image of the SG is inserted at the top right of the full-body section. **B** anterior SG; **C** anterior middle SG; **D** middle SG; **E** posterior SG. Scale bars: **A** 400 μm; (**B–E**) 200 μm; inset images, 20 μm.

**Fig. 2 | Ultrastructure of the silk cocoon of *N. cornuta*. A** SEM image of a cocoon with sparse perforations (arrows). **B**, **C** Close-ups of cocoon perforation(s). **D** Arrangement of silk fibers on the outside of the cocoon. Scale bars: (**A**, **B**) 100 μm and (**C**, **D**) 10 μm.

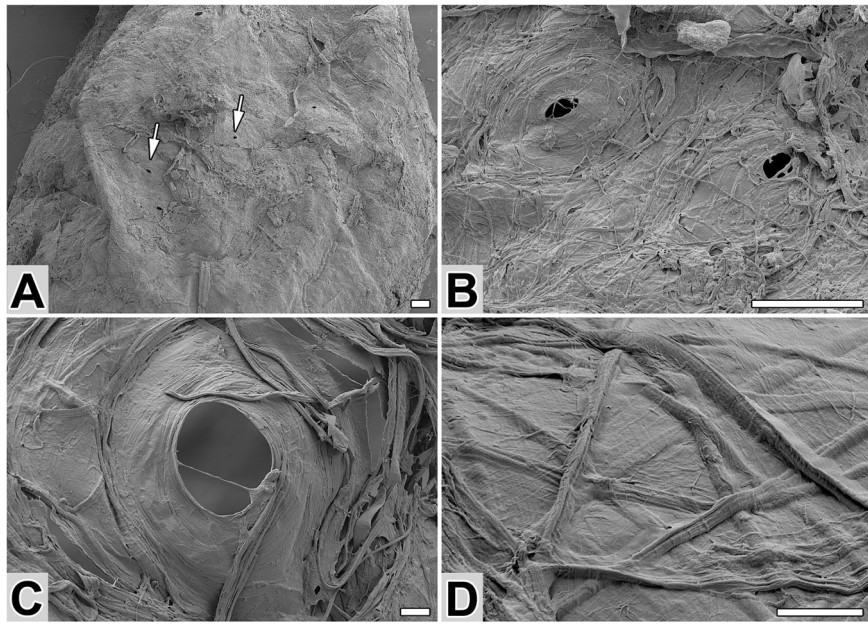

**Table 1 | List of putative structural proteins and their corresponding genes identified in the silk of *N. cornuta***

| Protein/gene | Contig | Exons | GenBank | Mw | No a.a. | pI | Gravy | aa1 | aa2 | aa3 |
|---|---|---|---|---|---|---|---|---|---|---|
| Sericin-like 1 | JAHKQU010000031.1 | 4 | BK070163 | 41226.27 | 398 | 5.69 | -0.885 | S 28.1 | L 11.1 | D 7.3 |
| Sericin-like 2 | JAHKQU010000031.1 | 2 | BK070164 | 933319.1 | 10692 | 12.37 | -0.67 | S 57.4 | G 15.8 | A 10.1 |
| Sericin-like 3 | JAHKQU010000031.1 | 2 | BK070161 | 1457778 | 13021 | 9.84 | -1.628 | N 23.6 | S 21.4 | T 10.8 |
| cadhezin-like 1 | JAHKQU010000021.1 | 2 | BK070165 | 134175.8 | 1206 | 7.48 | 0.219 | L 16.3 | I 9.7 | V 7.5 |
| cadhezin-like 2 | JAHKQU010000021.1 | 2 | BK070166 | 681274.4 | 6387 | 6.27 | 0.456 | L 16.7 | S 9.5 | I 9.1 |
| cadhezin-like 3 | JAHKQU010000101.1 | 2 | BK070173 | 66038.07 | 630 | 9.25 | -0.125 | G 14.1 | V 14.0 | P 12.4 |
| cadhezin-like 4 | JAHKQU010000101.1 | 2 | BK070173 | 66341.65 | 635 | 8.98 | -0.105 | G 15.0 | P 12.8 | V 12.4 |
| Zonadhesin-like A | JAHKQU010000021.1 | 5 | BK070169 | 27015.41 | 251 | 7.62 | -0.615 | C 15.1 | A,R 8.4 | S 8.0 |
| Zonadhesin-like B | JAHKQU010000021.1 | 19 | BK070162 | 115036.9 | 1053 | 6.14 | -0.574 | C 14.4 | P 8.3 | G 8.1 |
| Zonadhesin-like C | JAHKQU010000021.1 | 20 | BK070170 | 127501.5 | 1181 | 5.14 | -0.38 | C 14.6 | G 8.4 | N 7.5 |
| Zonadhesin-like D | JAHKQU010000021.1 | 6 | BK070171 | 26867.77 | 240 | 5.45 | -0.541 | C 14.6 | D,K 7.5 | A + S 5.8 |
| Mucin-like 1 | JAHKQU010000032.1 | 4 | BK070167 | 205416.5 | 1889 | 5.25 | -1.283 | T 15.1 | S 13.1 | G 8.9 |
| Mucin-like 2 | JAHKQU010000010.1 | 10 | BK070168 | 172033.6 | 1624 | 6.31 | -0.429 | G 12.3 | S 10.1 | Q 8.7 |

The table includes the GenBank contig number containing the gene of interest (from whole genome shotgun sequencing), the number of exons, and the GenBank accession number of the genomic sequence. Additionally, it provides the molecular weight of the predicted protein product (excluding the signal peptide), its isoelectric point, hydrophobicity (GRAVY score), and the three most abundant amino acids. Amino acid sequences of these proteins are shown in Fig. S4.

at its original position in *N. cornuta*, the region between *Pros* and the *DRSC* residue did not contain any sequence that could encode a larger repetitive protein such as FibH. Overall, *fibH* was absent in the synteny regions of *N. cornuta* and *M. aruncella*; however, it was present in the other species of both groups, supporting the hypothesis that the micropterigids incidentally lose their *fibH* gene.

**Possible *N. cornuta* ortholog of *fibL***

Neither FibH- nor FibL-like proteins were detected in the silk proteins of *N. cornuta*. To further investigate this, we employed the BLAST algorithm to thoroughly examine the transcriptome and genome sequences of *N. cornuta*, and the genome sequence of *M. aruncella* (based on the homology of terminal regions, and location on the contig as opposed to general similarity). We identified a distantly related sequence resembling *fibL*, named *fibX*, and organized it into six exons. When comparing the protein encoded by *fibX* with other FibL proteins from Lepidoptera and Trichoptera, we observed that the similarity was

markedly low (Fig. S1). The *fibX* protein shares only 25–28% identity with other *fibL* genes. Notably, we did not find any *fibL-like* gene in the *M. aruncella* genome.

To assess the conservation of synteny in the genomic regions harboring *fibL* in Lepidoptera and Trichoptera, we analyzed the corresponding sequences in several representatives of both taxa. We examined the genomes of four moths and one caddisfly (*B. mori, I. masculella, M. aruncella, N. cornuta*, and *Himalopsyche kuldschensis*). Our analysis revealed that in the species studied, the *fibL* gene is on the same contig/chromosome as several conserved genes, including dynein regulatory complex subunit 4 (*DRC4*) and histone deacetylase 11 (*HDAC11*). However, the order of these genes is not conserved, preventing us from identifying the exact location of FibL in micropterigids. Notably, the *fibX* gene in *N. cornuta* is located on the same contig as *DRC4* and *HDAC11*, suggesting that *fibX* could be an ortholog of *fibL*, which has undergone significant divergence. Additional data from micropterigids are needed to confirm that *fibX* is an ortholog of *fibL*.

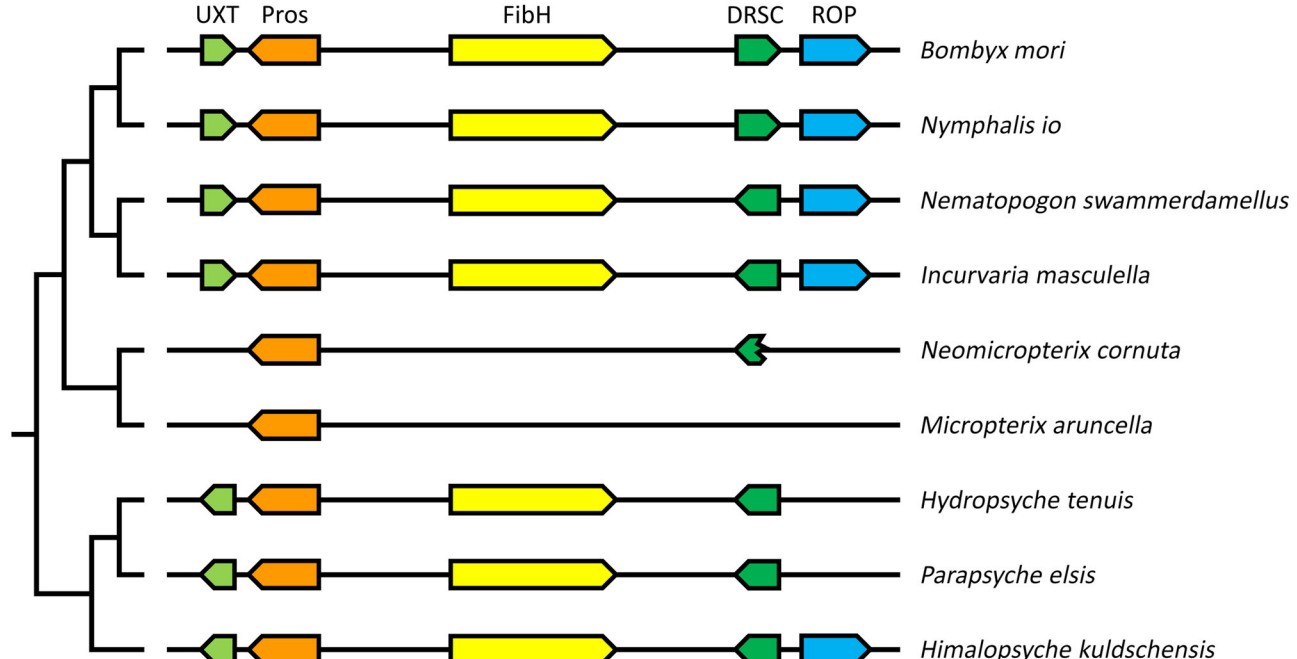

**Fig. 3 | Conserved synteny analysis of *FibH* and surrounding genes.** Genes and their relative position and orientation on chromosomes or scaffolds are indicated by block arrows. The same color indicates homologous genes. Homologs of *fibH*-related genes are shown for four other Lepidoptera and three Trichoptera species. Localization of the region in the represented species: *B. mori* – chromosome 25 (NC_051382.1); *N. io* – chromosome 19 (NC_065906.1); *N. swammerdamellus* – chromosome 20 (OX336353.1); *I. masculella* – chromosome 19 (OX335659.1); *N. cornuta* – contig JAHKQU010000018.1; *M. aruncella* – chromosome Z (OX155980.1); *H. tenuis* – contig VTON01000064.1; *P. elsis* – contig JAGVSN010000042.1; *H. kuldschensis* – contig JAHFWI010000014.1. Abbreviations of the surrounding genes encoding: *UXT* - Ubiquitously expressed Transcript protein; *Pros* - Homeobox protein prospero; *DRSC* - Dihydrolipoyllysine-residue succinyltransferase component of 2-oxoglutarate dehydrogenase complex; *ROP* - Repetitive organellar protein. *B. mori*: *UXT* - XM_062676197.1; *Pros* - XM_062676169.1; *FibH* - NM_001113262.1; *DRSC* - XM_012688929.4; *ROP* - XM_004921836.5. *Nymphalis io*: *UXT* - XM_050498294.1; *Pros* - XM_050498138.1; *FibH* - XM_050498483.1; *DRSC* - XM_050498484.1; *ROP* - XM_050498112.1.

## Loss of FibH and the loss or divergence of FibL is specific for Microptergidae

Is the absence of FibH specific to the family Micropterigidae or does it also apply to other Lepidopterans? For the subsequent analysis, we chose two representative species from the family Eriocraniidae in the suborder Eriocranoidea, which is characterized by its basal split from other Lepidopterans[15]. We generated transcriptomes of the final-instar larvae of both Eriocraniidae species and performed BLAST searches to identify homologous sequences of *fibH* and *fibL* genes. We successfully identified sequences similar to those of *fibH* and *fibL* in both eriocraniid species. The resulting alignments of putative FibH and FibL proteins with sequences of several homologs in both Lepidoptera and Trichoptera are shown in Figs. 4 and S1, respectively. Both proteins contain conserved sequences that are characteristic of fibroins. Our results suggest that the absence of *fibH* and substantial divergence or loss of *fibL* are specific to Micropterigidae.

## Discussion

Despite the contrasting habitats of terrestrial moth larvae and aquatic caddisfly larvae, both Lepidoptera and Trichoptera are believed to share a basic silk structure[13]. Silk in both groups comprises fibroin proteins (FibH and FibL), forming a core surrounded by a heterogeneous group of adhesive coat proteins[5]. However, various sources of evidence suggest that FibH may have been incidentally lost in the silk produced by micropterigid moth larvae. First, *N. cornuta* cocoons did not contain any fibroin proteins, as revealed by the proteomic analysis. Second, neither the transcriptome nor the genomic sequence of this species showed sequences closely related to *fibH*; only one sequence distantly related to *fibL* was found. Conserved synteny analysis of *fibH* and the surrounding genes revealed that the region of the *fibH* gene was absent in the examined micropterigid species. The results of proteomic, transcriptomic, and genomic analyses corroborated the insights from the morpho-histological observations. The liquid silk stored within SG of *N. cornuta* was only a single layer without distinct core and coat protein layers, and the spun silk was amorphous, resembling that synthesized by the fibroin mutant *Nd-s* of *B. mori* (Fig. 5)[19,20].

The absence of conserved *fibH* and *fibL* sequences in micropterigids, coupled with their presence in caddisflies[13] and all other examined moths from the suborders Eriocranoidea and Adeloidea, supports the hypothesis that *fibH* is likely secondarily lost in Micropterigidae while remaining conserved in other Lepidopterans. Consistently, both heavy and light fibroin chains have been previously found in *Phymatopus californicus*, a member of the family Hepialidae (17), and similar sequences have been also detected in *N. swammerdamellus* (family Adelidae, https://wellcomeopenresearch.org/articles/8-531). In the two studied micropterigid species, *fibL* was either absent or highly divergent. The closest sequence identified in *N. cornuta*, designated *fibX*, showed only 26.8% identity with *fibL* from *E. semipurpurella*. In contrast, the corresponding sequence in the caddisfly *P. conspersa* showed 39.09% identity with *E. semipurpurella fibL*. Additionally, a homolog of *fibX* was not detected in *M. aruncella*. Phylogenetic analysis indicated that *fibX* occupied an uncertain position in the cladogram because of its greater divergence (Fig. S2). We propose that FibX diverged after losing its original role as a FibH-binding partner.

These changes in the composition of silk molecules affect the morphology and mechanical properties of cocoons. Species with a higher proportion of fibrillar proteins tend to produce mesh-like silk structures, whereas those with more adhesive proteins form compact continuous layers[21]. *N. cornuta* cocoons are compact, smooth, and amorphous (Fig. 5A). These features are comparable to those of *G. mellonella* cocoons[7]. Likewise, the fibroin-deficient (*naked pupa*) mutant of *B. mori* creates compact, smooth cocoons with tightly bound fibers that are almost devoid of pores[20], similar to the dense, sericin-rich cocoons of African moths such as *Argema mimosae* and *Gonometa postica*[22]. This structure, with tiny openings, may facilitate gas exchange or water removal from the cocoon shells and thus

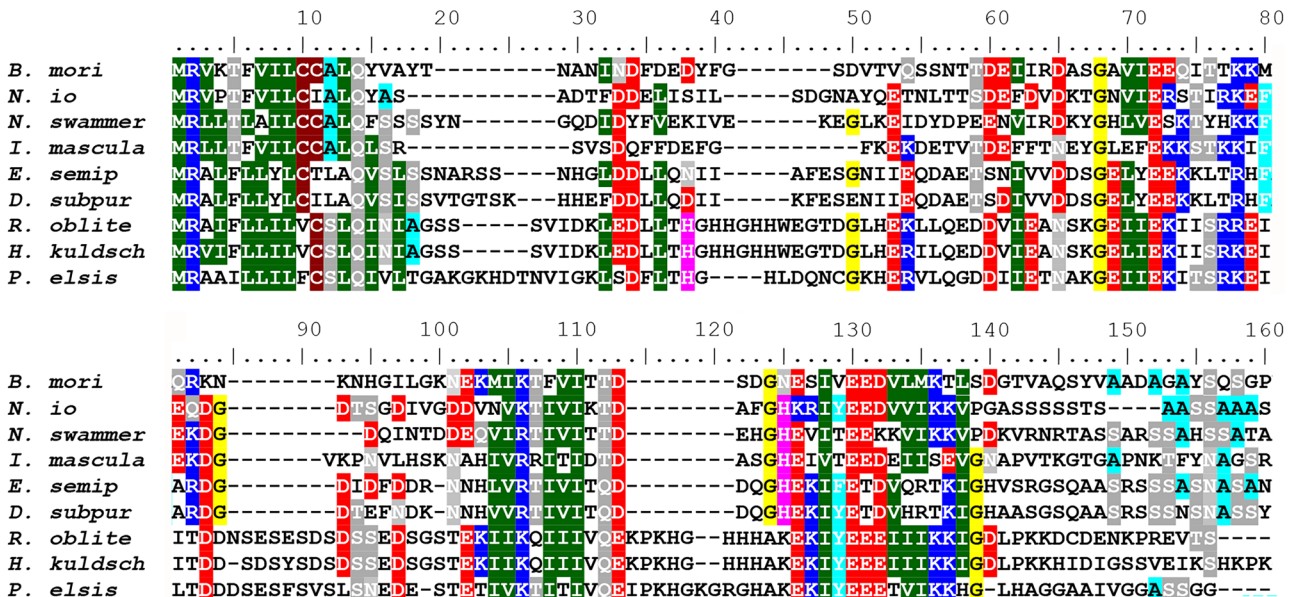

**Fig. 4 | Conservation of N-terminal FibH sequences between Lepidoptera and Trichoptera.** The species include seven lepidopterans (*Bombyx mori*, *Nymphalis io*, *Nematopogon swammerdamellus*, *Incurvaria masculella*, *Eriocrania semipurpurella*, and *Dyseriocrania subpurpurella*) and three trichopterans (*Hydropsyche tenuis*, *Parapsyche elsis* and *Himalopsyche kuldschensis*). The accession numbers of the FibH proteins are listed in the legend in Fig. 3; the accession numbers of *E. semipurpurella* and *D. subpurpurella* are PQ040461 and PQ040462, respectively.

may be suitable for the highly humid, typical pupation sites of micropterigid larvae.

Another possible consequence of the loss of fibroin in the micropterigid larvae is the degeneration of spinnerets, a distinctive trait in this family among Amphiesmenoptera (Fig. S3)[23,24]. Spinnerets are crucial for the final stages of silk formation in silk-producing insects as they apply shear forces essential for converting fibroin into functional silk[17,25]. Without FibH, the role of the spinneret becomes redundant, potentially leading to gradual degeneration and gradual disappearance.

As the micropterigid moths can still produce cocoons despite lacking fibroins, this challenges the conventional view that the FibH and FibL are highly conserved in the Amphiesmenopteran genomes and play a crucial role in the functionality of their silk. Notably, from the perspective of functional redundancy, they contain several large proteins with putative structural functions, termed as sericin- and cadhezin-like proteins instead of FibH. The amino acid sequences of these proteins are shown in Fig. S4. These proteins have features similar to those of pseudo-fibroins or cadhezins[9,10] concerning an exon-intron structure with a typical short first exon and a long second exon with repetitive sequences. Similar to pseudofibroins or cadhezins, *N. cornuta* cadhezin-like proteins lack the conserved N- and C-terminal regions typical of FibH molecules. In addition, four cadhezin-like proteins, including proline-rich Cazl 3 and 4 and slightly hydrophobic Cazl 1 and 2, may play a structural role. Sericin-like proteins 1, 2, and 3 in *N. cornuta* are extremely hydrophilic and may be involved in adhesion and/or water uptake. The examined micropterigid species possessed sericin- and cadhezin-like proteins, but their sequences differed significantly. Such differences may be related to the significant evolutionary divergence between these species and the distinct environments wherein these species pupate: under the moss for *N. cornuta* and in the upper soil layer for *M. aureatella*[26,27]. Overall, the presence of unique large silk protein sequences similar to cadhezins and pseudofibroins in aquatic caddisfly larvae may be related to ecological changes in their pupation sites in semiaquatic environments prone to frequent submersion.

The *N. cornuta* SG primarily comprises large secretory cells encircling the gland lumen. These secretory cells characteristic of PSG and are specialized to produce large amounts of insoluble fibroin in Lepidoptera and both soluble and insoluble proteins in Trichoptera. In contrast, MSG in

moths comprises a thin layer of smaller cells that are primarily responsible for the production of soluble coat proteins. As the MSG in *N. cornuta* is relatively short when compared to other Lepidoptera species, it can be assumed that the production of coat proteins is limited, as most proteins may be produced in the PSG. Although a thin surface layer is visible in Fig. 6 in the transmission electron microscopy images, we cannot definitively determine the origin of this layer within the MSG.

As shown in Fig. 5G–I, the silk structure at the breaking points shows that the inner part of the *N. cornuta* silk fiber is more porous than that of *B. mori*. Additionally, as shown in Fig. 6, *N. cornuta* silk exhibits a highly amorphous, putty-like character, which is in contrast with the more rigid structure of wild type and even *Nd-s* mutant silkworm silk, which retains its shape. We hypothesize that this amorphous nature is advantageous for the formation of cocoons with volcano-shaped openings, as it provides flexibility and adaptability in cocoon construction.

Additional research may uncover other factors contributing to silk protein diversity. Investigating a broader range of species could provide a more comprehensive understanding of the evolutionary divergence and functional redundancy of silk proteins in Amphiesmenoptera. Moreover, functional assays are needed to confirm functional redundancy and structural roles for sericin- and cadhezin-like proteins.

To the best of our knowledge, this study demonstrates the first naturally occurring case of silk production in Lepidopteran species within the Amphiesmenoptera, achieved without FibH and FibL. The proximate cause is secondary *fibH* gene loss, which is potentially compensated by the function of alternative genes. The unique positioning of Micropterigidae among Amphiesmenoptera in terms of morphology, ecology, and evolutionary history led to this discovery. Exploring the universal functions and redundancy of FibH alongside other proteins and unraveling the adaptive evolution and diversification of these gene clusters and their silk products can provide further insights when compared across diverse lineages representing deep nodes.

## Material and Methods
### Biological material
*N. cornuta* final-instar larvae were collected from the mountains of the Shikoku Island at two sites in Tosacho, Kochi Prefecture, Japan. The

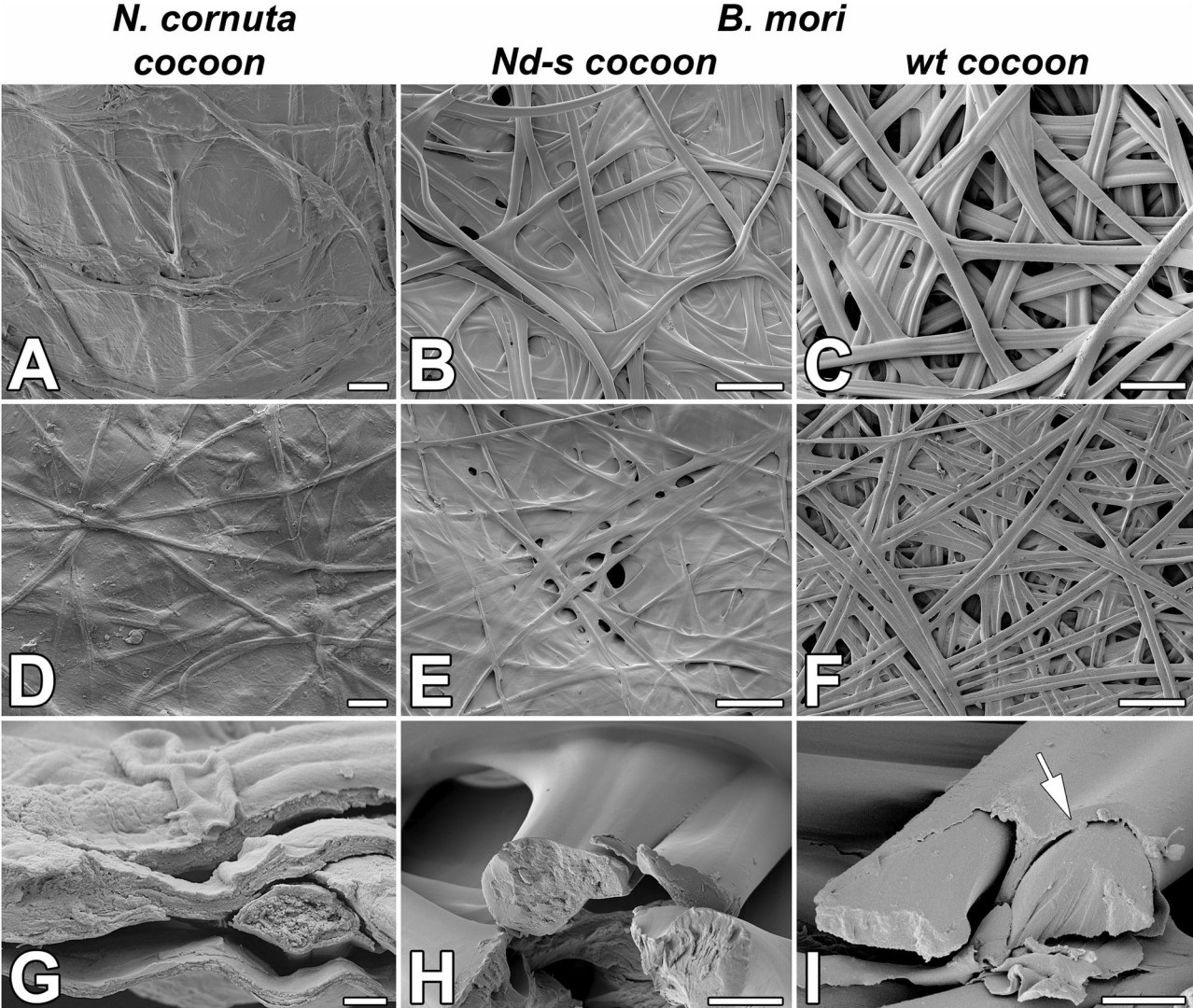

**Fig. 5 | SEM images of the silk of *N. cornuta* and *B. mori*.** Structure of cocoon and individual silk filaments of *N. cornuta* **A**, **D**, **G** *B. mori* mutant in FibL (*Nd-s* cocoon; **B**, **E**, **H**), and control *B. mori* (wt cocoon; C, F, I). **A–C** Scanning electron microscope (SEM) images of the outer side of the cocoon. **D–F**; SEM images of the inner side of the cocoon. **G-I** SEM images of broken fibers. Arrows point to the layer of sericin on the surface of *B. mori* wild-type silk fibers. The structure of the silk at the breaking points shows that the inner part of the silk fiber of *N. cornuta* is more porous than that of the silkworm. Scale bars: (**A**, **D**) 10 μm; (**B**, **C**, **E**, **F**) 100 μm; (**G**) 1 μm; (**H**, **I**) 10 μm.

specimens were pricked with a fine needle and stored in RNA later (Sigma-Aldrich) for several days before RNA isolation. *Eriocrania semipurpurella* and *Dyseriocrania subpurpurella* larvae were collected in Ceske Budejovice, Czech Republic.

### Micro-CT

Samples were fixed in Bouin-Hollande solution (BHS) for several days at 4 °C without acetic acid, but supplemented with mercuric chloride[28] The fixative was then washed in phosphate-buffered saline (PBS). The fixative was then washed in PBS and distilled water and contrasted with Lugol's solution for 14 days. The samples were scanned with high spatial resolution on a microCT SkyScan model 1272 (Bruker microCT, Land). A total of 1822 projections were acquired with a 360° rotation in 0.1°/ 0.1° rotation steps without filters. The micro-CT scans were performed using the following parameters: Voltage = 40 kV, current = 200 μA, the distance between object and source = 35.2 mm, the distance between detector and source = 275.3 mm, pixel binning = 2 × 2, and total scan duration = 2 h 40 min. Post-processing of the tomographic data: Cross-sectional slices were reconstructed using the volumetric NRecon reconstruction software SkyScan, version 2.2.0.6 (Bruker microCT, Belgium). A 3D model of the SG

in the context of the entire larva was created using Imaris software (version 10.0; Oxford Instrument, UK) – Surpass – Contour Surface Module.

### Histology and microscopy

Paraffin sections of *N. cornuta* bodies were labeled with Masson's trichrome. The cuticles of $CO_2$-anesthetized larvae were punctured with a fine needle using 4% paraformaldehyde as a fixative. Samples were fixed overnight at 4 °C and then rinsed in PBS (thrice for 15 min). Standard histological procedures were used for tissue dehydration, embedding in Paraplast, sectioning (10 μm), deparaffinization, and rehydration. Sections were washed in distilled water and stained with the HT15 Trichrome Staining Kit (Masson) (Sigma-Aldrich, Inc., St. Louis, MO, USA), according to the manufacturer's protocol. Stained sections were dehydrated and embedded in a DPX-embedding medium (Fluka). High-resolution images of the sections were acquired using a BX63 microscope, DP74 CMOS camera, and CellSens software (Olympus Corporation, Tokyo, Japan) by stitching multiple images and Z-stack imaging.

For transmission electron microscopy: the dehydrated cocoon samples were embedded in resin (Epon) by gradually increasing the volume ratio of resin to acetone (1:2, 1:1, and 2:1, each for 1 h at 20 °C). The samples were

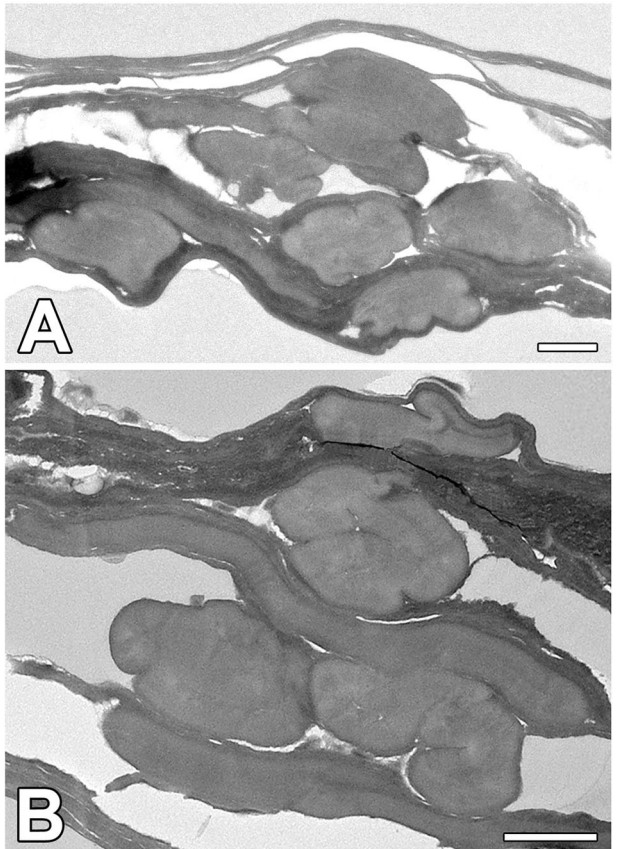

**Fig. 6 | TEM images of the cross-section of the fiber samples. A, B** Cross-section through the silk fibers of *N. cornuta*. Note: the amorphous character of silk fibers. (Scale bars: 1 μm).

incubated in resin for 24 h (20 °C) and then polymerized at 62 °C for 48 h. Ultrathin sections were cut on Leica ultramicrotome, stained with uranyl acetate-lead citrate, and observed using a JEM-1400 JEOL transmission electron microscope (Jeol, Akishima, Japan).

### Ultrastructure of silk

Silk samples were cut from cocoons, glued to the surface of aluminum holders, sputter-coated with gold, and analyzed using a JEOL JSM-7401F scanning electron microscope (Jeol, Akishima, Japan).

### Transcriptome preparation

Total RNA was isolated from the entire larval body of *N. cornuta, E. semipurpurella* and *D. subpurpurella* at the final-instar stage using TRIzol reagent (Invitrogen, Carlsbad, CA, USA)[29]. RNA sequencing libraries were prepared by NEBNext® Ultra™ II Directional RNA Library Prep Kit for Illumina (NEB #E7765, New England Biolabs, UK). The cDNA library was sequenced on a 2 × 150 bp Illumina platform (paired-end reads) using a NextSeq 500 sequencer. The reads of *N. cornuta* were mapped to the genome using RNA Star, and the transcriptome was constructed using BRAKER3 (Galaxy Version 3.0.6+ galaxy2). The de novo transcriptome assembly of *E. semipurpurella* and *D. subpurpurella* was performed using the Trinity software (v. 2.9.1) on the Galaxy platform[30]. The de novo and genome-guided transcriptomes resulted in 135,524 and 34,384 assembled contigs, respectively. The quality of the transcriptome assemblies was checked using BUSCO[31], using the Insecta odb10 dataset, revealing completeness of 98 and 96%, respectively. The raw data have been deposited in NCBI under the BioProject accession number: PRJNA1215287. Silk gland transcriptome of *Neomicropteryx cornuta* was deposited in the Dryad repository (https://doi.org/10.5061/dryad.4mw6m90n3).

### Protein identification using mass spectrometry

The cocoon proteins were dissolved in 8 M urea and processed according to Hughes et al.[32]. Briefly, the samples were washed, digested with trypsin, and acidified with trifluoroacetic acid to a final concentration of 1%. The peptides were then desalted using in-house C18 disk-packed tips (Empore, Oxford, USA)[33]. The processed samples were analyzed by nanoscale liquid chromatography coupled with tandem mass spectrometry (nLC-MS/MS). The resulting MS/MS spectra were compared with the *N. cornuta* protein database generated from the transcriptome[34].

### Synteny analysis

For synteny analysis, we used a local BLAST search of the genomic sequences of *B. mori, N. swammerdamellus, N. io, I. masculella, H. tenuis, P. elsis, H. kuldschensis, N. cornuta*, and *M. aruncella* using the BioEdit software[35]. We examined the genomic region surrounding the *B. mori* fibroin gene on chromosome 25 and identified several conserved genes: *Ubiquitously expressed transcript* (*UXT*), *prospero* (*Pros*), *dihydrolipoyllysine-residue succinyltransferase component of the 2-oxoglutarate dehydrogenase complex* (*DRSC*), and *repetitive organellar protein* (*ROP*). The BLAST algorithm was configured to search for homologous sequences with a minimum identity threshold of 80% and a minimum alignment length of 50 base pairs. Plots showing the microsyntenic relationships were then generated based on the best mutual hits between the species.

### Phylogenetic analysis

The relationship between the putative fibroin gene from *N. cornuta* (*fibX*) and other *fibL*s genes was also explored using a phylogenetic approach with cDNA sequences from four Lepidoptera and five Trichoptera species. Identities were calculated using the Ident and Sim modules of the Sequence Manipulation Suite[36]

A basic dendrogram (p-distance, neighbor-joining method) was constructed in MEGA[37], whereas a more rigorous phylogram (Maximum Likelihood algorithm) was built using the online version of IQ-Tree[38] (http://iqtree.cibiv.univie.ac.at/) with model selected by ModelFinder[39] and reliability assessed with ultra-fast bootstrap with 1000 replicates. Graphical editing of the trees was performed in tvBOT[40].

### Reporting summary

Further information on research design is available in the Nature Portfolio Reporting Summary linked to this article.

### Data availability

The experimental data supporting the results of this study are available in this article or in the supplementary materials. The raw data have been deposited in NCBI under the bioproject accession number PRJNA1215287. Silk gland transcriptome of *Neomicropteryx cornuta* was deposited on the Dryad repository. https://doi.org/10.5061/dryad.4mw6m90n3. The reported nucleotide sequence data are also available in the Third Party Annotation Section of the DDBJ/ENA/GenBank databases published recently[41] under the accession numbers TPA: BK070161-BK070173.

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

## Acknowledgements

This research was supported by European Community's Program Interreg Bayern Tschechische Republik BYCZ01-039. We would also like to thank Mgr. Pavel Talacko (BIOCEV—Biotechnology and Bio-medicine Centre of the Academy of Sciences and Charles University, funded by the European Regional Development Fund—CZ.1.05/1.1.00/02.0109) for his great help with proteomic analyses and the core facility Laboratory of Electron Microscopy, Biology Centre CAS supported by the MEYS CR (LM2023050 Czech-BioImaging) and ERDF (No. Z.02.1.01/0.0/0.0/16_013/0001775). Yume Imada was supported by a research grant for Environmental Field Research by the Asahi Glass Foundation (Asahi Glass Co., Ltd.), Grant-in-Aid for Scientific Research (KAKENHI) grant numbers JP20K15852 and JP23K23947 from the Japan Society for the Promotion of Science (JSPS). Miluše Hradilová

was supported by RVO:68378050, ELIXIR CZ research infrastructure project (MEYS Grant No: LM2023055) including access to computing and storage facilities. We thank Jitka Pflegerová for technical support and Gabriela Krejčová (Department of Molecular Biology and Genetics, Faculty of Science, University of South Bohemia in České Budějovice) for help with microCT. We would also like to thank Dr. Hidetoshi Teramoto (NARO, Tsukuba, Japan) for *B. mori Nd-s* cocoons.

## Author contributions

Y.I. and Mi.Z. conceptualized the work, developed the methodology, and planned the experiments. Mi.Z. and L.R. analyzed the *N. cornuta* transcriptome, Y.I. and L.R. collected insect material, H.S., and S.P. performed micro-CT analysis, histochemistry, and electron microscopy, M.H. prepared cDNA libraries, Ma.Z. performed phylogenetic analysis. Mi.Z. wrote the manuscript with input from all authors.

## Competing interests

The authors declare no competing interests.
