## [Transparent Peer Review file · Communications Biology]

Absence of fibroin H sequences and a significant divergence in the putative fibroin L homolog in *Neomicropteryx cornuta* (Micropterigidae) silk.

Corresponding Author: Professor Michal Zurovec

Version 0:

Reviewer comments:

Reviewer #1

(Remarks to the Author)

In this study, the authors identified the silk gland RNA and cocoon proteins of *N. cornuta* through RNA-seq and LC-MS, discovering that the cocoon of *N. cornuta* is distinct from other Lepidopteran cocoons, as it lacks both the fibroin heavy chain and the light chain, which is a noteworthy finding. Additionally, they conducted analyses on the morphology and histology of the *N. cornuta* silk gland, the ultrastructure of the silk cocoon of *N. cornuta*, as well as FibH and the surrounding genes. However, the study did not delve deeply into the mechanisms behind the cocoon's formation, and therefore, I believe that the research is not yet at the level required for publication in *Communication Biology*.

Major comments

1. The authors have identified and classified the RNA and proteins in the silk glands and cocoons of *N. cornuta* using RNA-seq and LC-MS. However, they were unable to distinguish in which segments of the silk gland (ASG, MSG, and PSG) the silk proteins were synthesized. Although the cocoon lacks FibH and FibL, it contains unique proteins that may function similarly to FibH as structural proteins. Understanding the synthesis of these proteins in specific silk gland segments could potentially explain how silk proteins can form a cocoon without FibH.

2. Morphologically, the cocoons of *N. cornuta* and *Nd-s* appear similar (Fig. 5). In fact, there is a significant difference in the protein composition of their cocoons, which is likely to result in substantial differences in their microstructures.

Characterizing the structure and protein distribution of the cocoons using a variety of methods would aid in understanding how a cocoon can form without FibH. Therefore, I believe it is necessary for the authors to provide additional experimental data.

Minor comments

1. The notation for FibH and FibL is inconsistent, as seen in Line 31 and Line 32: FibH, FibL; and Line 37: fibH, fibL. Please check the entire document for consistency.

2. The citation of references in the main text is not standardized; some are superscript while others are not, as seen in Line 44: 1; Line 51: (6), and Line 64: (13, 18). Please carefully check the entire document.

3. In Line 79, under the first subheading of the results, "SG" should be written out in full, not abbreviated.

4. Line 129-130, the contig numbers should not be italicized. Please check the entire document carefully.

5. In Fig. 1E, the magnified image at the top right of the full-body section does not exactly match the silk gland indicated by the arrow in the original figure.

Reviewer #2

(Remarks to the Author)

A nicely written and convincing study.

I was confused by In 280 pg 7 which implied that the silk production in this species occurred without FibH and highly divergent FibL. The wording implies that it does contain highly divergent FibL (which table 1 shows it does not). I suggest

that the authors remove the 'highly divergent' from this sentence as the silk does not appear to contain any FibL, highly divergent form or not.

Other minor comments

Define CT (micro-CT)

In the 'morphology of SG and cocoons' section, when the authors say that various sections of the SG correspond to regions in other moths and caddisflies, what is this correspondence based on?

Pg 5, ln 178: presumably FibH should be FibL, please change

Pg 5 ln 195: please either explain what additional data from micropterigids would confirm that these are orthologous or remove this sentence.

Pg 5 ln 199 please reword the first sentence – it is well established that Fib H is found in many Lepidoptera. Consider changing to ...'does it apply to other related Lepidoptera?'

Pg 6 ln 233 Please reword 'The closest sequence....' Consider changing to "the DNA with the closest sequence...'

Reviewer #3

(Remarks to the Author)

Manuscript Background

The authors of this study produce a characterization of silk genes/proteins in *Neomicropteryx cornuta*, a moth in the group Micropterigidae, which is sister to all other Lepidoptera. Understanding the silk protein profile for this group could provide great evolutionary insight into the gain and loss of important silk genes used in lepidopteran life histories. This will be the first characterization of a non-ditrysian lepidopteran which has secondarily lost an integral silk protein (FibH), giving important understanding to silk use of basal lepidopterans. The authors of this study take a well-rounded approach to characterizing the loss of FibH by integrating synteny of silk genes between several members Amphiesmenoptera, using proteomics to characterize protein identity in silk products, computed tomography (CT scanning) to internally characterize silk-organ structure, and RNAseq for further identifying lack of FibH. In addition to this important find, the authors describe a putative homolog for FibL, with divergence that has not been previously recorded. Both of these proteins are highly influential of silk phenotype in Lepidoptera, and their findings provide novel insights to the field. The evidence provided for this characterization is beyond what several current studies use, and serves as a model for characterizing genomic and phenotypic structures for silk. I fully support this paper moving forward to publication, given the minor edits attached below.

Comments to Authors

1) In the end of the introduction paragraph (70-76), I would add a sentence to emphasize the importance of characterizing the Micropterigidae silk proteins as part of the knowledge gap mentioned (ie. how it's mentioned in the abstract that this group is sister to all other silk-producing lepidoptera, and can provide evolutionary insights to an important biomaterial used in lepidopteran life histories). This is emphasized well in the abstract, but is missing in the introduction.

2) In line 90, I would also add a sentence about what would be needed to clearly distinguish segments of the SG (ie. HCR methods to look at gene expression/lack of in FibH or adhesive genes). May not be necessary since FibH is lost here, and FibL is highly diverged. But would be useful to orient the reader by saying that these designations are based on tissue-specific expression, and in order to designate different segments, it would be beneficial to have expression information, in addition to histology color differences.

3) Would denote here (line 178) that the lack of similarity to FibH and FibL is based on the homology of terminal regions, and location on the contig as opposed to general similarity. Earlier in the text you mention that the RNA results from *N. cornuta* had repetitive silk-like proteins, but it may be worth clarifying that these are not like FibH since it's also repetitive.

4) Line 207, this is an incredible find! I would however mention the existing losses of FibL in other lepidopterans (some Saturniidae, like *Actias luna* do not have FibL). This line just makes it seem like FibL loss is specific to Micropterigidae. I think a good solution for this would be to mention the other groups which FibL is lost, and highlight that this extreme divergence of FibL in Micropterigidae is unique to the group. Or say that the loss of FibL in basal Lepidoptera is unique to Micropterigidae.

5) Line 297, would clarify what reagent BHS fixative is, maybe full name or catalog number. I would do the same with the PBS solution, and add "iodine" after Lugol's. The methods outlined for the proceeding section, in regards to the detail of your reagents, is good to model here.

Version 1:

Reviewer comments:

Reviewer #3

(Remarks to the Author)

The authors of this study present a characterization of the silk genes present in the moth *Neomicropteryx cornuta* (Family: Micropterigidae), which is a group hypothesized to have split from all other extant Lepidoptera approximately 300mya.

Understanding the evolution of the silk proteins in this group could provide great evolutionary insight to silk production for Amphiesmenoptera, and the gains and losses of genes vital to lepidopteran life histories. To my knowledge, this article will be the first characterization of a non-ditrysian lepidopteran which shows the loss of heavy chain fibroin (FibH). The evidence provided by the authors support their hypotheses of gene loss, and their genomic and morphological characterization for silk in this species takes a well-rounded approach by including protein identification in silk products, computed tomography (CT scanning) to internally characterize silk-organ structure, and RNAseq for further identifying lack of FibH. In addition to this find, the authors describe a putative homolog for FibL, with divergence that has not been previously recorded. Both of these proteins are highly influential of silk phenotype in Lepidoptera, and this study provides novel insights to the field. The evidence provided for this characterization is beyond what several current studies use.

Answers to reviewers' comments

Reviewer #1

*1. The authors have identified and classified the RNA and proteins in the silk glands and cocoons of *N. cornuta* using RNA-seq and LC-MS. However, they were unable to distinguish in which segments of the silk gland (ASG, MSG, and PSG) the silk proteins were synthesized. Although the cocoon lacks FibH and FibL, it contains unique proteins that may function similarly to FibH as structural proteins. Understanding the synthesis of these proteins in specific silk gland segments could potentially explain how silk proteins can form a cocoon without FibH.*

Response: Thank you for your comment. Unfortunately, it is extremely difficult to obtain a large quantity of *N. cornuta* larvae, which limits our ability to conduct a more comprehensive investigation. However, our histochemical analysis of the silk gland (SG) reveals distinct differences in cell morphology between the posterior silk gland (PSG) and middle silk gland (MSG) segments, consistent with observations in other Lepidoptera and Trichoptera. We have added following text to the discussion: "*N. cornuta* SG primarily comprises large secretory cells encircling the gland lumen. These secretory cells are characteristic of PSG and are specialized to produce large amounts of insoluble fibroin in Lepidoptera and both soluble and insoluble proteins in Trichoptera. In contrast, MSG in moths comprises a thin layer of smaller cells that are primarily responsible for the production of soluble coat proteins. As the MSG in *N. cornuta* is relatively short when compared to other Lepidoptera species, it can be assumed that the production of coat proteins is limited, as most proteins may be produced in the PSG. Although a thin surface layer is visible in Fig 6 in transmission electron microscopy images, we cannot definitively determine the origin of this layer within the MSG".

*2. Morphologically, the cocoons of *N. cornuta* and *B. mori* Nd-s mutant appear similar (Fig. 5). In fact, there is a significant difference in the protein composition of their cocoons, which is likely to result in substantial differences in their microstructures. Characterizing the structure and protein distribution of the cocoons using a variety of methods would aid in understanding how a cocoon can form without FibH. Therefore, I believe it is necessary for the authors to provide additional experimental data.*

Response: Thank you for your comment. We prepared new samples of the silk fibers of *N. cornuta* and *B. mori* for both scanning and transmission electron microscopy. We added following text in the manuscript to the discussion: "As shown in Figure 5d–f, the silk structure at the breaking points reveals that the inner part of the *N. cornuta* silk fiber is more porous than that of *B. mori*. Additionally, as shown in Figure 6, *N. cornuta* silk exhibits a highly amorphous, putty-like character, which is in contrast with the more rigid structure of wild type and even Nd-s mutant silkworm silk, which retains its shape. We hypothesize that this amorphous nature is advantageous for the formation of cocoons with volcano-shaped openings, as it provides flexibility and adaptability in cocoon construction."

3. The notation for FibH and FibL is inconsistent, as seen in Line 31 and Line 32: FibH, FibL; and Line 37: fibH, fibL. Please check the entire document for consistency.

Response: Thank you for your comment. We have added „protein“ orthologs of FibH and FibL for clarity.

4. The citation of references in the main text is not standardized; some are superscript while others are not, as seen in Line 44: 1; Line 51: (6), and Line 64: (13, 18). Please carefully check the entire document.

Response: Thank you for your comment. We have corrected the citations as suggested.

5. In Line 79, under the first subheading of the results, "SG" should be written out in full, not abbreviated.

Response: Thank you for your comment. We have adjusted the text as suggested.

6. Line 129-130, the contig numbers should not be italicized. Please check the entire document carefully.

Response: Thank you for your comment. We have corrected the text as suggested.

7. In Fig. 1E, the magnified image at the top right of the full-body section does not exactly match the silk gland indicated by the arrow in the original figure.

Response: Thank you for your comment. We have selected the best images of the entire body section and a detailed view of the SG. We have revised the figure legend to clarify that the insertion is not an enlargement of the total section, but only a highly magnified image.

Reviewer #2

8. I was confused by ln 280 pg 7 which implied that the silk production in this species occurred without FibH and highly divergent FibL. The wording implies that it does contain highly divergent FibL (which table 1 shows it does not). I suggest that the authors remove the 'highly divergent' from this sentence as the silk does not appear to contain any FibL, highly divergent form or not.

Response: Thank you for your comment. We have detected the transcript of putative fibL, but not the protein. We have corrected the text as suggested.

9. Define CT (micro-CT)

Response: Thank you for your comment. We have added an explanation that micro-CT imaging is a technique that uses X-rays to view the inside of an organ layer by layer.

10. In the 'morphology of SG and cocoons' section, when the authors say that various sections of the SG correspond to regions in other moths and caddisflies, what is this correspondence based on?

Response: Thank you for your comment. The morphology of the SG cells is similar to other lepidopteran species. The posterior part of the SG consists of large secretory cells with large nuclei that produce most of the secretory proteins. The adjacent middle part, as in living lepidopterans, consists of flat cells covering the glandular lumen. We have adapted the text on page 3 accordingly.

11. Pg 5, ln 178: presumably *FibH* should be *FibL*, please change

Response: Thank you for your comment. We are sorry, we did not find any mention of *FibH* in this section.

12. Pg 5 ln 195: please either explain what additional data from micropterigids would confirm that these are orthologous or remove this sentence.

Response: Thank you for your comment. We have removed the sentence as suggested.

13. Pg 5 ln 199 please reword the first sentence – it is well established that *Fib H* is found in many Lepidoptera. Consider changing to ... 'does it apply to other related Lepidoptera?'

Response: Thank you for your comment. We have adjusted the sentence as suggested.

14. Pg 6 ln 233 Please reword 'The closest sequence....' Consider changing to "the DNA with the closest sequence...'

Response: Thank you for your comment. We have adjusted the sentence as suggested.

Reviewer #3

15. In the end of the introduction paragraph (70-76), I would add a sentence to emphasize the importance of characterizing the Micropterigidae silk proteins as part of the knowledge gap mentioned (ie. how it's mentioned in the abstract that this group is sister to all other silk-producing lepidoptera, and can provide evolutionary insights to an important biomaterial used in lepidopteran life histories). This is emphasized well in the abstract, but is missing in the introduction.

Response: Thank you for your comment. We have added the sentence „can provide evolutionary insight to an important biomaterial in lepidopteran life history“ as suggested.

16. In line 90, I would also add a sentence about what would be needed to clearly distinguish segments of the SG (ie. HCR methods to look at gene expression/lack of in *FibH* or adhesive

genes). *May not be necessary since FibH is lost here, and FibL is highly diverged. But would be useful to orient the reader by saying that these designations are based on tissue-specific expression, and in order to designate different segments, it would be beneficial to have expression information, in addition to histology color differences.*

Response: Thank you for your comment. We agree, however, as we mentioned in the explanation to the comment 1 of the first reviewer, it is extremely difficult to obtain a large quantity of *N. cornuta* larvae.

17. Would denote here (line 178) that the lack of similarity to FibH and FibL is based on the homology of terminal regions, and location on the contig as opposed to general similarity. Earlier in the text you mention that the RNA results from N. cornuta had repetitive silk-like proteins, but it may be worth clarifying that these are not like FibH since it's also repetitive.

Response: Thank you for your comment. We have added the sentence clarifying the homology searches as suggested.

18. Line 207, this is an incredible find! I would however mention the existing losses of FibL in other lepidopterans (some Saturniidae, like Actias luna do not have FibL). This line just makes it seem like FibL loss is specific to Micropterigidae. I think a good solution for this would be to mention the other groups in which FibL is lost, and highlight that this extreme divergence of FibL in Micropterigidae is unique to the group. Or say that the loss of FibL in basal Lepidoptera is unique to Micropterigidae.

Response: Thank you for your comment. We have added the explanation that the loss of FibL also occurred in moths of the family Saturniidae in which the FibH molecules form dimers connected with a disulfide bridge. The loss of FibH and the extreme divergence or loss of FibL is ,however, unique to Micropterigidae.

19. Line 297, would clarify what reagent BHS fixative is, maybe full name or catalog number. I would do the same with the PBS solution, and add "iodine" after Lugol's. The methods outlined for the proceeding section, in regards to the detail of your reagents, is good to model here.

Response: Thank you for your comment. We have explained that BHS is Bouin-Hollande fixation solution and added a citation (Levine et al., 1995).